# Molecular Mechanism of Cannabinoids in Cancer Progression

**DOI:** 10.3390/ijms22073680

**Published:** 2021-04-01

**Authors:** Cristina Pagano, Giovanna Navarra, Laura Coppola, Maurizio Bifulco, Chiara Laezza

**Affiliations:** 1Department of Molecular Medicine and Medical Biotechnology, University of Naples “Federico II”, Via Pansini 5, 80131 Naples, Italy; pagano.cris@gmail.com (C.P.); vanna.navarra@libero.it (G.N.); coppola.laura6@gmail.com (L.C.); 2Institute of Endocrinology and Experimental Oncology, IEOS CNR, Via Pansini 5, 80131 Naples, Italy

**Keywords:** cannabinoids, cell death, migration, angiogenesis, cancer

## Abstract

Cannabinoids are a family of heterogeneous compounds that mostly interact with receptors eliciting several physiological effects both in the central and peripheral nervous systems and in peripheral organs. They exert anticancer action by modulating signaling pathways involved in cancer progression; furthermore, the effects induced by their use depend on both the type of tumor and their action on the components of the endocannabinoid system. This review will explore the mechanism of action of the cannabinoids in signaling pathways involved in cancer proliferation, neovascularisation, migration, invasion, metastasis, and tumor angiogenesis.

## 1. Introduction

Endocannabinoids are endogenous lipid, synthesized from phospholipids of the cell membrane. They modulate a variety of physiological processes through the activation of G-protein coupled CB1 (cannabinoid1) and CB2 (cannabinoid2) receptors. Endogenous cannabinoids, receptors, and the enzymes involved in the synthesis, transport, and degradation of endocannabinoids constitute the endocannabinoid system (ECS). The main endocannabinoids are AEA (N-arachidonoylethanolamine), and 2-AG (2-arachidonoylglycerol), derived from arachidonic acid. Anandamide is produced in a calcium-dependent manner, then released by neurons following depolarization and inactivated by the membrane-bound enzyme serine hydrolase fatty acid amide hydrolase (FAAH). AEA binds to both cannabinoid receptors, showing greater efficacy at CB1 than CB2 receptors. 2-AG, isolated in 1995, is present at high concentrations in the central nervous system. 2-AG is synthetized by the activation of phospholipase C (PLC) and diacylglycerol lipase (DAGL), and is hydrolyzed to arachidonic acid and glycerol in the mouse brain by monoacylglycerol lipase; MAGL. 2-AG binds both CB1R and CB2R with great efficiency [1]. CB1R and CB2R are differently distributed in the tissues and have different expression levels. Both CB1 and CB2 cannabinoid receptors are associated with the Gi/o family of G proteins, which inhibit adenylyl cyclase activity and enhance extracellular signal-regulated protein kinase activity [2]. CB1R has been discovered thanks to Δ9-tetrahydrocannabinol (THC) and identified in the brain [3]. It is abundant in the hippocampus, basal ganglia, cerebellum, and prefrontal cortex modulating several physiological functions like emotions, motor activity, cognition, appetite, and memory. CB1Rs are also present in the uterus, testes, ovaries, and prostate [4]. CB2Rs are localized in tissues of the immune system, involved in immunological functions, and expressed on astrocytes and microglia in the central nervous system [5]. However, they have also been revealed in several rat brain areas (cerebellum and hippocampus) and peripheral tissues like adipose, skeletal muscle, and endocrine pancreas [6]. Recent pharmacological studies described the existence of putative orphan cannabinoid receptors, the non-CB1 non-CB2 receptors, which have been identified by observing AEA’s ability to stimulate pharmacological effects in both endothelial cells and the brain of transgenic mice lacking the endocannabinoid receptors. AEA, in particular, interacts with serotonin and muscarinic receptors and with the transient receptor potential vanilloid type-1 (TRPV1) receptor belonging to the family of the six-transmembrane-domain transient receptor potential (TRP) channels. 2-AG, on the other hand, is weakly active on TRPV1 [3]. Others orphan GPCR receptors (oGPCR), putative non-CB1/CB2 receptors, have been discovered in the vasculature, brain, and immune cells like GPR55 and GPR18 receptors. GPR55 modulates immune/inflammatory processes. GPR18 is expressed in the brainstem, hypothalamus, testis, spleen, and is localized in the cellular membrane and in the macrophages and microglia so to modulate the downstream signaling [7]. Finally, it has been described that AEA and 2-AG activate the peroxisome proliferator-activated receptors (PPAR)-α and -γ, nuclear receptors, thus controlling the genes expression implicated in metabolism and inflammatory responses [3]. Another enzyme responsible for the degradation of AEA, OEA (oleoylethanolamide), and PEA (palmitoylethanolamide) is the lysosomal hydrolase N-acylethanolamine-hydrolyzing acid amidase (NAAA) [3].

## 2. ECS in Cancer

### 2.1. CB Receptor Expression in Cancer

Currently, numerous publications deal with the involvement of the endocannabinoid system in tumor progression [8]. Alterations of CB1R and CB2R expression levels and/or function have been observed in cancer; this also holds true for ECS enzymes and concentration of endogenous cannabinoids. Alterations of cannabinoid receptors have been observed in glioblastoma, and cells arising from tumor samples. Elevated CB2R expression correlated with a higher degree of tumor malignancy. Results relating to the CB1Rs are conflicting. Some authors revealed a decrease, others no alteration and/or an increase in the CB1R expression in high-grade glioma in comparison with low grade and the healthy brain [8]. In breast, prostate, and pancreatic cancer, elevated CB2R levels correlated with the malignancy grade of the cancer. In breast cancers, high expression of CB1R and CB2R was also described in human breast tumor biopsies [9,10]. Moreover, elevated CB2R expression correlated with tumor malignancy. Hormone receptor-negative-tumors showed a CB2R higher expression and have a favorable prognosis [11]. In biopsies of patients with endometrial cancer, CB2Rs were up-regulated, instead, other studies highlight low levels of both receptors [12]. Furthermore, high levels of CB1R were found in invasive ovarian tumors [13]. CB1R was elevated in lung carcinomas, in 24% (7 of 29) of cases, and CB2R in 55% (16 of 29) [14]. Malignant thyroid lesions showed elevated CB1R and CB2R expression. In particular, elevated levels of CB2R correlated with the presence of metastases in the lymph node, and with the greatest risk of cancer recurrence [11]. In prostate carcinoma, the higher expression of CB1Rs is correlated with a higher degree of malignancy and a greater probability of giving rise to metastases. About cancers of the digestive tract, several studies show a link between the CB1R overexpression and cancer prognosis, e.g., in patients with colon adenocarcinoma whereas the elevated expression of CB2Rs caused the highest proliferative levels and lymph node involvement. More, in ApcMin/+ mice, the knockout of CB1R stimulated the growth of intestinal adenoma, suggesting a suppressive role of CB1R [15]. In the aforementioned mice, CB1R was more expressed in inflamed tissue than in tumor tissue, while GPR55 expression has been observed to be inversely regulated, acting oppositely to CB1R [11].

In human prostate cancer cell lines (LNCaP, DU145, PC-3, CWR22Rν1), high levels of CB1R and CB2R expression were detected compared with non-cancerous prostate cells and PZ-HPV-7 cells [16]. The GPR55 receptor was highly expressed in MDA-MB-231 cells compared to MCF-7 breast cancer cell lines [17]. The GPR55, GPR18, and GPR119 receptors were observed in hCMEC/D3 human brain endothelial cells and A2058 melanoma cells. GPR55 was highly expressed in Hodgkin’s lymphoma and non-Hodgkin’s lymphoma cell lines. Moreover, a high CB1 receptor expression was also found in Hodgkin lymphoma cells [18].

### 2.2. Endocannabinoid Levels and Degrading Enzymes in Cancer

The AEA and 2-AG levels are altered in tumor biopsies compared to normal tissues. Brain cancer had lower AEA levels [19,20], while other data revealed higher levels in glioblastomas and meningiomas [21] comparing to normal tissues. In gliomas, the expression pattern and activities of NAPE-PLD (N-acylphosphatidylethanolamine-specific phospholipase D), FAAH (fatty acid amide hydrolase), and MAGL are decreased. The AEA downregulation in glioma tissues was correlated with the decrease of activity and expression of NAPE-PLD, the enzyme responsible for AEA synthesis and with a reduced the activity and expression of FAAH, enzyme responsible of AEA degradation. The increase of 2-AG levels in glioma tissues correlated with a decrease of the activity and expression of MAGL, the 2-AG degrading enzyme, while the expression of DAGL-α, the enzyme involved in the 2-AG synthesis, was unchanged [20]. In colon carcinoma and adenomatous polyps, high levels of AEA and 2-AG were revealed [22,23]. In breast cancer, AEA levels were not increased, unlike its precursor, N-acylphosphatidylethanolamine, which was expressed at high levels [24]. Interestingly, AEA levels were elevated in lymphatic metastasis [25] and in endometrial cancer (EC) tissue compared to non-cancerous tissues. Moreover, FAAH was reduced while NAPE-PLD was increased in EC [26]. In another study, NAPE-PLD and MAGL expression was upregulated in colorectal cancer tissue [25,27,28], furthermore, high MAGL expression was indicative of a poor prognosis for patients compared to those with a low expression of MAGL [29]. In addition, levels of both AEA and 2-AG were elevated in human pituitary adenomas compared with non-cancerous tissues. Endocannabinoid levels have been increased in CB1 receptor-positive samples, while they were lower in samples with a low or absent CB1R expression [30]. Prostatic biopsies analyses show a higher expression of FAAH protein in adenocarcinomas in comparison with non-cancerous tissue [31]. High MAGL expression was observed in ovarian tumors, in colorectal cancer tissues [27,28,32], and ductal breast tumors [24]. Longer survival of pancreatic ductal adenocarcinoma patients correlated with elevated levels of FAAH and MAGL [33]. In endometrial carcinoma, the downregulation of MAGL compared to healthy tissue was observed [34].

## 3. Cannabinoid Ligands

Cannabinoids are diverse hydrophobic compounds divided into three categories: (1) endocannabinoids, (2) phytocannabinoids, and (3) synthetic cannabinoids.

### 3.1. Endocannabinoids

At present, other putative endogenous CB1 and CB2 receptor ligands as 2-arachidonoyl-glyceryl ether or noladin ether (2-AGE), O-arachidonoyl-ethanolamine (virodhamine), N-arachidonoyl-dopamine (NADA), and oleamide (OA) have been found to have anticancer effects. NADA has been observed to reduce prostate carcinoma cell proliferation via nuclear factor (NF)-κB/cyclin D- and cyclin E-dependent pathways [35], and it inhibited the invasion of prostate cancer cells by downregulation of protein kinase A (PKA) activity [36]. NADA has been shown to inhibit breast cancer cell growth in vitro and in vivo [37], in human osteosarcoma, lymphoma, and leukemia cell lines [38] and in colorectal carcinoma cells via activation of CB1R [23], while the inhibition of cell growth of neuroblastoma cells was mediated by TRPV1 receptor activation [39]. NADA inhibited plasma membrane translocation and neoplastic transformation of oncogenic KRAS4A redistributing the cytoplasmic NRAS to the Golgi apparatus [40]. In contrast, there is a limited effect of virodhamine on tumor progression.

### 3.2. Phytocannabinoids

Gaoni and Mechoulam discovered the main compound of the *Cannabis sativa* plant, the Δ9-tetrahydrocannabinol (Δ9-THC), progenitor of the phytocannabinoid family [41]. It carries out its action via two G protein-coupled receptors CB1R and CB2R. Through CB1R, it produces the psychoactive effects of marijuana [42]. More than 150 compounds have been isolated. The main drugs are Δ9-THC, cannabidiol CBD, cannabinol CBN, and cannabichromene (CBC), followed by ∆8-THC, cannabidiolic acid (CBDA), cannabidivarin (CBDV), and cannabigerol (CBG), and are collectively known as phytocannabinoids [42]. Δ9-THC (THC) has a robust anti-cancer potential. It can reduce the growth of breast cancer cells. In contrast, some authors found that THC can stimulate breast cancer cell proliferation [43]. THC treatment, in a mouse model of ErbB2-driven metastatic breast cancer, decreased tumor proliferation, as well as lung metastases. Additionally, it caused apoptosis and inhibited angiogenesis [9]. In glioma cell lines, THC caused inhibition of cell viability and cell death by autophagy; moreover, it decreased glioma tumor growth in vivo. THC can inhibit the cell growth in vitro and in vivo of many tumors, like myeloma, leukemia, melanoma, and hepatocellular carcinoma. [43]. In recent years, cannabidiol, in particular, has aroused intense research for its therapeutic efficacy for neurological disorders, alone or in combination with THC [44]. Nabiximols (Sativex), a standardized extract of THC and cannabidiol has been approved in some countries to mitigate spasticity in patients with multiple sclerosis [45]. However, in the clinic, nabiximols have been used as analgesic for the treatment of cancer pain [46]. CBD exhibits a lower CB1R and CB2R affinity compared to THC; it acts as an antagonist to CB1R in the mouse brain membranes [47], and displays inverse agonism to human CB2R. CBD may interact with TRPVs, 5-HT1A, GPR55, and PPARγ. Moreover, it has anti-cancer effects by inducing apoptosis and inhibiting cell migration, and metastasis in different cancer types [48]. The combined administration of THC and CBD inhibited the proliferation of glioblastoma cells and in mouse models of cancer. Temozolomide, in combination with THC, and/or THC + CBD (at a 1:1 ratio), exerts a strong anti-tumoral action in glioma xenografts [49]. Cannabigerol, O-1602, and URB-602 have demonstrated an antineoplastic action in experimental models reducing tumor volume and ACF formation [50]. The cannabichromene (CBC) has anti-proliferative action in prostate cell lines, in colorectal cancer Caco-2 cells, and MDA-MB-231 and MCF-7 breast cancer cell lines. Cannabidivarin (CBVD) and cannabinol (CBN) have pro-apoptotic effects on human prostate cell lines. CBVD inhibits the growth of colon cancer cells while CBN does so in breast cancer cells [43]. Cannabidiolic acid (CBDA) inhibited the invasivity of breast cancer cells and downregulated the c-fos and the COX-2 [51].

### 3.3. Synthetic Cannabinoids

Some synthetic cannabinoids have been used as pharmacological tools for the study of the ECS in cancer progression. They include: (a) inhibitors of endocannabinoid cellular uptake, such as AM404, VDM11, UCM707, OMDM-2; (b) inhibitors of FAAH, such as URB-597, and N-arachidonoyl-serotonin, AA-5H which also antagonizes TRPV1 receptors; (c) inhibitors of MAGL, such as URB602 and JZL184; (d) dual CB1/CB2 agonists, such as WIN-55,512-2, CP-55940, and HU-210; (e) anandamide analogs that are more metabolically stable, such as methanandamide and metfluoroanandamide; (f) selective CB1 agonists, such as arachidonoylchloroethanolamide (ACEA) and arachidonoylcyclopropylamide (ACPA); (g) selective CB2 agonists, such as HU-308, JWH-015, JWH-133; (h) selective antagonists/inverse agonists for CB1Rs, such as SR141716A (rimonabant), AM251 and AM281; (i) selective CB2 antagonist/inverse agonists such as SR144528, AM630. It is important to note that the above-mentioned compounds have been used in experimental models of cancer in which the cannabinoids were shown to have antiproliferative effects on tumor progression [3].

#### 3.3.1. Inhibitors of Endocannabinoid Cellular Uptake

The anti-bacterial AM404 targeted colon carcinoma cells by suppressing the oncogenic E3 ligase FBXL5, to inhibit differentiation and migration CRC cells [52]. AM404, VDM11, UCM707, and OMDM2 have antiproliferative action on C6 glioma cells dependent on cell density. VDM-11, a selective anadamide uptake inhibitor, and arachidonoyl-serotonin (AA-5-HT), a blocker of endocannabinoids enzymatic hydrolysis, inhibited the cell proliferation in vivo and in vitro of rat thyroid transformed (KiMol) cells [53]. OMDM-2 by itself exhibited antiproliferative effects on both MCF-7 breast cancer cells and U-87 glioma cells [54].

#### 3.3.2. Inhibitors FAAH

Two FAAH inhibitors arachidonoyl serotonin (AA-5HT) and URB597 inhibited metastasis of A549 lung cancer cells due to the overexpression of tissue inhibitor of matrix metalloproteinases-1 (TIMP-1) [55]. URB597 in association with palmitoylethanolamine (PEA) inhibited the viability of B16 melanoma cells. This combination led to reduced melanoma progression in vivo [56]. These drugs in combination with Met-F-AEA down-regulated cyclin D1 and CDK4 expressions and caused apoptosis via activation of caspase-9 and PARP in non-small cell lung cancer. This treatment inhibited cancer growth in a xenograft nude mouse model [57]. U3B597 in combination with AEA decreased cell proliferation in N1E-115 Neuroblastoma cells [58]. Finally, URB597 inhibited the cell proliferation of colon cancer DLD1 cells and in SW620 cells [59].

#### 3.3.3. MAGL Inhibitor

The MAGL inhibitor URB-602 in combination with Cannabigerol and O-1602, induced apoptosis, inhibited angiogenesis, and reduced tumor volume and ACF formation on CRC models in vivo [50]. URB602 inhibited colon cell growth and angiogenesis down-regulating VEGF and FGF-2, and decreased azoxymethane-induced preneoplastic lesions, polyps, and tumors [28]. JZL184 attenuated tumor proliferation and metastasis of the PC3 prostate cancer cells in mice [60]; it increased apoptosis in colorectal cancer cell lines increasing cell sensitivity to 5-fluorouracil [61], suppressed hepatocellular carcinoma growth and progression in vivo [62], and increased survival rate in the xenograft mouse model by blocking PGE2 synthesis [63]. Furthermore, JZL184 decreased the progression of bone metastasis, skeletal tumor proliferation, and osteolysis in mouse models of osteosarcoma and prostate or breast cancers [64].

#### 3.3.4. Dual CB1R/CB2R Agonists

WIN 55,212-2 inhibited tumor proliferation in prostate cancer via CB2R [65,66] and in human BEL7402 hepatocellular carcinoma cells [67]. WIN 55,212-2 and CP 55-940 induced cell death in C6 (rat) and U373 (human) glioma tumour lines [68]. Moreover, CP55940 selectively induced cell apoptosis in Jurkat cells and in whole bone marrow CD3+ cells from 3 T-ALL patients [69]. HU 210 reduced mouse P19 embryonal carcinoma (EC) cells viability [70] and inhibited breast cancer and prostate cancer cell proliferation [17,71].

#### 3.3.5. Anandamide Analogs

R (+)-Methanandamide had pro-apoptotic effects in human cervical cancer cells mediated by COX2 [72] and exerted anti-growth effects in prostate cells (PC-3) [73]. Metfluoroanandamide inhibited the growth of MDA-MB-231 cell line and in combination with URB597 reduced cell growth in neuroblastoma, lung, and colon cancer, or in combination with PEA in melanoma cancer cells [11].

#### 3.3.6. Selective CB1R Agonists

ACEA and JWH-133 inhibited the release of VEGF-A and VEGF-C factors by human lung-resident macrophage. In addition, ACEA inhibited the invasivity of breast cancer stem cells [74]. The synthetic cannabinoid arachidonylcyclopropylamide (ACPA) induced autophagic cell death by inducing ROS production in pancreatic cancer cells [75].

#### 3.3.7. Selective CB2R Agonists

JWH-015 had an anti-proliferative effect in several cancer cells, such as PC-3 prostate cancer cells, metastatic breast cancer MCF-7 cells, and lung cancer cell lines [76]. It inhibited the activation of EGFR signaling transduction pathway in ERα-cells in vitro and in vivo [77]. JWH-015 may block the secretion of factors by M2 macrophages cultured with A549 cells and inhibit their recruitment at the tumor site [78]. JWH-133 stopped cell proliferation and migration of the glioma cell and breast cancer cells in vitro and in vivo [11,79]. It decreased the transendothelial migration rate of melanoma cells [80]. JWH-133 in combination with ACEA inhibited LPS-induced synthesis of VEGF-A, VEGF-C, and angiopoietins and affected IL-6 secretion [81].

#### 3.3.8. Selective Antagonists/Inverse Agonists for CB1 Receptors

SR141716A inhibited carcinoma colon cell growth and decreased the number of neoplastic lesions in a colon mouse model [82]. It blocked glioma cell growth and induced cell apoptosis. SR141716 increased the expression of MICA/B levels, elevating glioma cell recognition by NK-cells [83]. It inhibited cell proliferation of breast cell lines via a mechanism mediated by lipid rafts [84]. AM251 induced pro-apoptotic effects in carcinoma pancreatic and colon cancer cells. Additionally, AM251 with ACEA decreased the invasivity of cancer stem cells [11]. AM251 significantly decreased tumor cell growth, induced apoptosis, and reduced migration of the renal carcinoma cell lines [85,86].

#### 3.3.9. Selective Antagonists/Inverse Agonists for CB2 Receptors

The best known CB2-selective antagonists/inverse agonists are diarylpyrazole (SR144528) and 6-iodopravadoline (AM-630) [87]. These compounds have been used as CB2 receptor antagonists; however, some studies report the ability of SR144528 to inhibit breast cancer cells proliferation by acting as a agonist [88]. AM630 displays anti-proliferative and anti-migratory effects on renal cell carcinoma [89].

## 4. Mechanism of Action of the Cannabinoids on Cancer Progression

### 4.1. Cell Cycle

Regulation of the cell cycle is an important phase to the survival of a cell, including the detection and repair of genetic damage and the control of correct cell division. Loss of this cell-cycle control is a hallmark of cancer cells, therefore targeting the cell cycle pathway is a mighty strategy in cancer therapy. Several studies have demonstrated that cannabinoids mediated cell cycle dysregulation and inhibition of proliferation in cancer cells. WIN 55,212-2 inhibited cell cycle progression in prostate cancer cells, PC3, and DU145 cells, causing an accumulation of cells in the G1 phase and decrease of cells in the S phase of the cell cycle, through the CB2R. In particular, WIN-treated cells showed a significant increase in the expression of p27 and a decrease of Cdk4, suggesting that cell cycle dysregulation is mediated through Cdk inhibitory subunit in the G0-G1 phase. It was demonstrated that the WIN treatment involves the p27-Cdk4-pRb pathway and the downregulation of Cdk4 resulted in the phosphorylation and inactivation of pRb, raising the suggestion that cannabinoids could arrest the cell cycle progression via inhibition of transcription factors necessary for the progression of cells through the G1 phase [65]. In SGC-7901 cells, human gastric cancer cell line, the cannabidiol (CBD), an allosteric negative modulator of CB1R and CB2R activity, stopped cell cycle in phase G0-G1 and significantly increased the expression levels of protein ataxia telangiectasia-mutated gene (ATM) and p53 and reduced p21, CDK2, and Cyclin E protein levels [90]. THC, through activation of CB2R, also reduced cell cycle progression in HS-and HR-breast cancer cell lines, determining cell cycle stops at the G2-M phase, by reduction of Cdc2 and induction of ROS synthesis, provoking cell death [91]. THC effect on cell cycle progression, alone or in combination with CBD, in multiple myeloma cell lines, U266 and RPMI8226 (RPMI) MM cell lines, has been also evaluated. THC stopped cells in the G1 phase after 24-h post-treatment, accompanied by increased proportions of sub-G1 phase cells after 48-h post-treatment, compared with their respective control. The THC-CBD combination was more effective in inducing the G1 cell population and the sub-G1 phase at 24-h post-treatment and in increasing cell accumulation in the sub-G1 phase at 48 h, compared with THC or CBD alone [92]. Furthermore, AEA inhibited cell cycle progression by arresting G1-S transition, as shown in the EFM-19 cells, the human prolactin-sensitive breast cell line [91]. In several human breast carcinoma cell lines, AEA, through CB1R activation, caused inhibition of cAMP synthesis and cell cycle arrest in the G1/S phase, while activation of CB2Rs by THC arrested the cells in the G2-M phase [93]. It was recently observed that AEA caused, by a receptor-independent mechanism, an arrest in the G2-M phase in St-T1b cells (human endometrial cell line) through the inhibition of the Akt pathway [94] (illustrated in Figure 1 and Table 1).

### 4.2. Apoptosis

Apoptosis is a form of programmed cell death that is essential for the development and survival of organisms [95]. Cannabinoids activate apoptosis either through CB1 or CB2 receptors. New pieces of evidence report that CBD promoted cell death in various gastric cancer cells AGS, MKN45, SUN638, and NCI-N87 cell lines. CBD induced apoptotic cell death by suppressing X-linked inhibitor apoptosis (XIAP) in a dose- and time-dependent manner. CBD significantly increased the ubiquitination of XIAP, which is regulated by Smac. Upon CBD treatment, Smac translocated from mitochondria in the cytosol and linked to the XIAP to increase its ubiquitination, increasing apoptosis. Moreover, CBD inhibited XIAP by stimulating endoplasmic reticulum (ER) stress-related genes in gastric cancer cells [95]. The CBD treatment of the SGC-7901 cell line increased the protein levels of cleaved caspase-3 and caspase-9, subsequently inducing apoptosis cell death. CBD increased Bax and decreased Bcl-2 expression levels, causing a reduction of the ratio of Bcl-2/Bax, which, in turn, determined an increase of mitochondrial membrane permeability and a decrease of mitochondrial transmembrane potential, thus allowing the release of cytochrome C into the cytosol [90]. CBD was shown to induce apoptosis in human HCT116 and DLD-1 cells through the cleavage of PARP and caspase-3, caspase-8, and caspase-9. Jeong et al. reported that Noxa, a pro-apoptotic member belonging to the Bcl-2 protein family, is important for CBD-induced apoptosis in CRC cell lines. CBD treatment increased Noxa in a dose- and time-dependent manner. Noxa caused ROS production, which further exacerbated the apoptosis. CBD induced ROS, a well-known inducer of (ER) stress, which is associated with Noxa activation [95]. THC, via CB1R activation, inhibited both PI3K/Akt and RAS-MAPK/ERK survival pathways in colorectal carcinoma cell lines, as well as in human Jurkat leukemia T cells, THC inhibited ERK, and Akt signaling. CBD inhibited the survival of both estrogen receptor-positive and estrogen receptor-negative breast cancer cell lines and induced apoptosis cell death in a concentration-dependent manner compared with MCF-10A cells, non-tumorigenic mammary cells. CBD induced apoptosis via the activation of caspase-8, the translocation of BID to the mitochondria, the release of cytochrome c and SMAC into the cytosol, and increased levels of Fas-L, suggesting the hypothesis that CBD induced cell death through the activation of the intrinsic apoptotic pathway in breast cancer cells. Furthermore, CBD enhanced ROS generation, suggesting that ROS plays a critical role in the CBD-induced apoptosis of breast cancer cells [96]. CBD with THC, CBG, CBN—this combination is called C6—induced cell cycle block in G2 phase and consequent apoptosis in the MCF-7 cell line. C6 significantly increased” Endoplasmic Reticulum Chaperone Protein Glucose-regulated Protein 78” (GRP78) levels, indicating that endoplasmic reticulum stress could cause both apoptosis and autophagy [97]. In prostate cancer cells, endocannabinoids AEA and 2-AG via CB1R induced apoptosis, causing an increase in the levels of activated caspase-3 and a reduction in the levels of Bcl-2. Finally, endocannabinoid treatment activated the Erk pathway and at the same time produced a decrease in the activation levels of the Akt pathway [94]. THC via activation of CB1R induced ceramide accumulation and Raf1/ERK activation in rat C6 glioma cells. More, the activation of CB2R by JWH-133 seemed to be a critical event leading to the inhibition of glioma growth in vivo. However, apoptosis by cannabinoids is not exclusively carried out by CBRs. AEA can induce apoptosis through TRPV1 activation led to oxidative stress, increasing calcium influx, and caspase activation [98]. R (+)-methanandamide (metAEA), a non-hydrolyzable AEA derivative, promoted apoptosis in human cervical carcinoma cells by inducing COX-2 expression and subsequent prostaglandins synthesis by activating PPARγ [72]. AEA, through GPR55 activation, induced apoptosis in cholangiocarcinoma cell lines, by the recruitment and activation of the death complex Fas/FasL [99]. AEA selectively induced ER stress-mediated apoptosis in non-melanoma skin cancer (NMSC) cells that overexpress COX-2. Specifically, the AEA cytotoxic effects were likely produced by the novel prostaglandin, 15d-PGJ2-EA, which was synthesized as a consequence of the metabolism of AEA by COX-2. Furthermore, apoptosis induced by AEA was also regulated by oxidative stress that was partially mediated by a reduction in total glutathione levels. Thus, cell death in NMSC cells is likely regulated by the conversion of AEA to 15d-PGJ2-EA, which diminishes intracellular glutathione levels leading to oxidative stress, ER stress, and ultimately apoptosis [94,100] (illustrated in Figure 1 and Table 1).

### 4.3. Autophagy

Autophagy is a mechanism that leads to the degradation of damaged cytoplasmic components. This is an old evolutionary process that involves the packaging of cell organelles by a two-membrane bag called an autophagosome. Cellular autophagy or autophagocytosis is a cellular mechanism of selective removal. Cannabinoids can induce autophagy in several cell lines and mouse models of cancer. The most important mechanism by which cannabinoids cause autophagy is the accumulation of ceramide in tumor cells. Cannabinoids increase ceramide concentration in the cell by two mechanisms. The first is the hydrolysis of sphingomyelin by the sphingomyelinase enzyme, thus creating ceramide only through activation of the CB1R. The second is de novo synthesis of ceramide with the enzyme serine-palmitoyl transferase (SPT), which generates ceramide by activating both CB1 and CB2 receptors. The accumulation of ceramide in the cell stimulates the stress response of the ER and the activation of different proteins CHOP, ATF-4, and p8, which promote the interaction of tribbles pseudokinase 3 (TRIB3) with Akt, causing the inhibition of the PI3K/AKT/mTOR pathway [101,102]. The signaling pathway p8/ATF4/CHOP/TRIB3, followed by the inhibition of the PI3K/AKT/mTOR cascade, is probably the most important antitumoral mechanism of cannabinoids. Another way to increase the ceramide levels by inhibiting the PI3K/AKT/mTOR pathway is the activation of Ca^2+^/calmodulin-dependent kinase kinase β (CaCMKKβ) activated by ER stress. The next step is the activation of AMPK, which directly phosphorylates and activates Tuberous Sclerosis Complex 2 (TSC2), the major direct inhibitor of mTORC1. Therefore, the inactivation of mTORC1 leads to increased autophagy. This mechanism has been observed in hepatocellular carcinoma cells and pancreatic carcinoma cells through activation of CB2R and induction of autophagy by AMPK [94]. Sativex^®^, a mixture of THC and CBD, activated CB1 and CB2 receptors by stimulating autophagy in a mouse model of tauopathy, associated with reduced levels of tau and amyloid proteins in the brain [94]. The THC-CBD treatment in multiple myeloma (MM) cell lines increased the sub-G1 cell accumulation due to an autophagic-cell death process. Indeed, after 24 h of treatment, these cells showed the conversion of the soluble form of LC3 (LC3-I) to form (LC3-II), a marker of autophagy activation. CBD alone induced a slight increase of LC3-II/LC3-I ratio, THC alone does not have an effect, while the THC-CBD combination strongly augmented the levels of the cleaved LC3-II form and the LC3-II/LC3-I ratio [92]. Detailed analysis indicated that THC treatment conduct to the formation of autophagosomes in human astrocytoma cell lines and primary cultures of human glioma cells. TRIB3 expression induced by THC promoted the interaction of the ER stress-related proteins with Akt decreasing its phosphorylation, as well as of its direct substrates TSC2 and PRAS40, which in turn resulted in mTORC1 inhibition [103]. THC-treatment decreased phosphorylation of p70S6 kinase (a well-established mTORC1 downstream target) and its substrate phospho-S6 ribosomal protein. Altogether, THC treatment triggered the following cascade of intracellular events: upregulation of ER-stress-related TRIB3, mTORC1 inhibition, and induction of autophagy, which led to apoptotic glioma cell death. These findings indicate that THC promotes cell death through stimulation of ER stress and autophagy in human glioma cells and is essential for cannabinoid anti-tumoral action in vivo [103]. The activation of CB1 receptors by AEA induced autophagy in a model of the mature intestinal epithelium, Caco-2 cell line, causing a reduction of the regulatory protein suppressor of cytokine signaling 3 (SOCS3) levels. Instead, in the spinal cord of mice with experimental autoimmune encephalomyelitis, the synthetic agonist HU-308 activated CB2 receptors and promoted autophagy, thereby leading to an inhibition of NLRP3 inflammasome activation, alleviating in this way the pathogenesis of this disease in this animal model [103] (illustrated in Figure 1 and Table 1).

### 4.4. Migration

Cell migration is a physiological process that also takes part in pathological processes, such as the migration of cancer cells and the development of metastases. Cannabinoids effects have been shown to have anticarcinogenic properties by inhibiting the migratory and invasive capacity of tumoral cells. WIN55,212-2 (WIN) can block the migratory capacity in osteosarcoma MG63 cells by reducing the activity and intracellular levels of the metalloproteases MMP2 and MMP9. This effect is independent of the secreted SPARC protein, which is acidic and rich in cysteine. SPARC is a matricellular protein important in tissue remodeling and extracellular matrix deposition. The release of SPARC is prevented by WIN; the latter, in fact, can retain the protein inside the cells by inhibiting the canonical secretory pathway upstream. Furthermore, WIN can also increase the release of extracellular vesicles containing miR-29b1, which is important for regulating cell proliferation and migration. Indeed, cells transfected with stable miRNA-29b1 showed reduced cell migration, similar to WIN-treated cells. In particular, it has been shown that WIN in osteosarcoma cells can influence cell migration in a miR-29b1-dependent manner and independently of the SPARC protein [104]. Several indole-based cannabinoid agonists, including WIN, interact with two additional targets: AI-sensitive GPCRs and microtubules. Six newly-developed indole-based compounds (ST-11, ST-23, ST-25, ST-29, ST-47, and ST-48) that exhibit distinct binding affinities at AI-sensitive receptors, CBRs, and the colchicine site of tubulin have shown inhibition of cells migration of the glioma mouse DBT cell line. Activation of AI-sensitive receptors with ST-11 inhibited both the basal and stimulated migration of the DBT mouse glioma cell line. The results obtained with ST-48 suggest that ST compounds regulate cell migration. Specifically, this agonist of the AI-sensitive receptors potently inhibits both basal cell motility and directed migration, and does not interact with tubulin. These small molecules acting at AI-sensitive receptors could reduce or stop the invasion of glioblastoma [105]. Cannabidiol (CBD) can induce cell death in multiple myeloma (MM). Cannabidiol treatment induced cell death also in combination with the proteasome inhibitor bortezomib. The treatment of CBD/THC in combination with other chemotherapeutic drugs acts in a synergic way, suggesting their use in combinatory therapy in several types of human cancers. A poor prognosis of breast cancer is correlated with high levels of metalloproteases. The CBD-THC combination was able to reduce cell migration of the MM cell line by downregulating expression of the chemokine receptor CXCR4 and the CD147 plasma membrane glycoprotein. Treatment with CBD and THC reduced expression of the β5i subunit and in synergy with carfilzomib (CFZ), a promising immuno-proteasome inhibitor that creates irreversible adducts with the β5i subunit, increased MM cell death, and inhibited cell migration [92]. THC and JWH-133 reduced tumor growth and the amount/severity of lung metastases in MMTV-neu mice. Cannabinoids modulated MMP activity via inhibition of MMP2 and activation of the MMP9 gene. Indeed, high levels of metalloprotease, amplification, or overexpression of the ERB2 gene have been found in the biopsies of breast tumors [9]. THC and Met-AEA decreased human cervical cancer (HeLa) cell invasion in a time- and concentration-dependent manner mediated by an increased expression of TIMP-1, which plays critical roles in the acquisition of migration and invasive ability by cancer cells [106]. Other authors have observed that THC administration downregulated TIMP-1 expression in mice bearing gliomas and also in two patients with recurrent glioblastoma multiforme [107]. In human breast cancer cells, inhibition of adhesion and migration, via decreased tyrosine phosphorylation of focal adhesion kinase (FAK), was observed after Met-F-AEA treatment [108]. In human non-small cell lung cancer A549 and SW-1573 cell lines, THC, via CB receptors, inhibited EGF-induced growth, chemotaxis, and chemoinvasion; moreover, THC inhibited lung cancer growth and metastasis in vivo murine model. THC reduced the EGF-induced phosphorylation of ERK1/2, JNK1/2, and AKT, responsible for the reduced migration and invasion observed in lung cancer cells [109]. G-protein coupled receptor 55 (GPR55), a lysophospholipid receptor, has been shown to play an important role in the migration and metastasis in colon cancer cells. In fact, in metastatic colon cancer cell line (HCT116) treatment with CID16020046, i.e., a selective GPR55 antagonist, or cannabidiol, showed a significant decrease in adhesion to endothelial cells and migration. In HCT116, the integrity of endothelial cell monolayers was increased after pre-treatment with the antagonists or after GPR55 siRNA knockdown while pre-treatment with lysophosphatidylinositol (LPI), the endogenous ligand of GPR55, decreased integrity of the monolayers [110]. Furthermore, the CB2R synthetic agonist JWH-133 and the CB1 and CB2 receptors agonist WIN inhibited cell proliferation and migration in vitro conditions. These results were confirmed in vivo in various mouse model systems. Cancer mouse models treated with JWH-133 or WIN showed a 40% to 50% reduction in tumor growth and a 65% to 80% reduction in lung metastasis. These effects were reversed by CB1 and CB2 receptor antagonists AM251 and SR144528, respectively, suggesting the involvement of both receptors [10] (illustrated in Figure 2 and Table 2).

### 4.5. Angiogenesis

The hallmark of cancer cells is the ability to proliferate, despite the absence of mitogenic signals. The growth of solid tumors is dependent on the generation of new vascular supply. Therefore, targeting neoangiogenesis constitutes one of the most promising therapeutic approaches against cancer. Cannabinoids, in cancer cell, can block the activation of the vascular endothelial growth factor (VEGF) pathway, which is an inducer of angiogenesis. Specifically, different molecules of this cascade, such as the main ligand (VEGF) and the active forms of its main receptors (VEGFR1 and VEGFR2), are downregulated in skin carcinomas, gliomas, and thyroid carcinomas after cannabinoids treatment [111]. Likewise, cannabinoids treatment decrease the formation of distant tumor masses in animal models and inhibit adhesion, migration, and invasiveness of glioma, breast, lung, and cervical cancer cells in vitro culture. These effects are dependent, at least in part, on the modulation of extracellular proteases (MMP2) [107] and their inhibitors such as tissue inhibitors of matrix metalloproteinases 1 (TIMP1) [112]. It has been observed that pharmacological inhibition of ceramide biosynthesis abolished the antitumor and anti-angiogenic effect of cannabinoids in glioma xenografts and reduced VEGF production by glioma cells in vitro and in vivo. Additionally, blocking ceramide biosynthesis and knocking-down of p8 gene prevented the inhibition of MMP2 expression induced by THC in glioma cell invasion. These observations suggested a general role of the ceramide and p8 regulated pathway in the antitumor activity of cannabinoids that target CB1 and CB2 receptors [107]. CBD, by acting independently of CB1 and CB2 receptors, produced an antitumor effect including the reduction of invasiveness and metastasis in different cancer animal models. This effect of CBD seems, at least partly, to depend on the downregulation of the helix–loop–helix transcription factor inhibitor of DNA binding 1 (ID1) [101]. HU-331 was able to reduce the area of the cells that proliferated from the aorta, the number of new vessels, and also their length. Indeed, after treatment with HU-331, the area occupied by the proliferating cells was much smaller, and both the length and number of vessels were reduced. The authors analyzed 96 key genes of major endothelial cell functions, including angiogenesis, vascular tone, adhesion, and cell injury after treatment with HU-331. The results showed that the levels of six genes such as matrix metalloproteinase-1, cyclooxygenase-2, and osteoprotegerin were increased, while the mRNA levels of the monocyte chemotactic protein-1, von Willebrand factor (VWF), and cytosolic phospholipase A2 were reduced. Other quinone cannabinoids such as HU-336 (tetrahydrocannabinol quinone) and HU-345 (cannabinol quinone) were also capable of inhibiting the aortic ring, but not compared to HU-331. Indeed THC and CBD were not only less potent than HU-331 but even at low concentrations (50nM) induced angiogenesis of the aortic ring [113]. N-arachidonoyl serine (ARA-S) is an endocannabinoid-like lipid with a weak affinity for the cannabinoid receptors (CB1 and CB2) and the transient receptor potential vanilloid receptor 1 (TRPV1). It enhanced angiogenesis in HUVEC incubated with different concentrations of ARA-S (0, 0.1, 1, 10 mM) compared to vehicle controls in vitro angiogenesis assays [114]. Moreover, ARA-S may show protective properties through stimulation of VEGF-C production and its cognate receptors (VEGFR-2 and 3) as well as through the GPR55 receptor activation. ARA-S-induced migration was significantly decreased in the GPR55 siRNA-transfected cells, as compared with the control siRNA-transfected cells [115] (illustrated in Figure 2 and Table 2).

### 4.6. Epithelial to Mesenchymal Transition (EMT)

Acquisition of invasive traits by tumor cells involve specific phenotypic changes associated with epithelial to mesenchymal transition (EMT), and regulated transdifferentiation process in which carcinoma cells lose cell-to-cell junctions and cell polarity and acquire migratory and invasive properties [116]. Cannabinoids (the active components of Cannabis sativa) and their derivatives have received considerable interest due to reports that they can affect tumor growth, migration, and metastasis. Several studies have shown that cannabinoid agonist WIN was involved in gastric cancer (GC) metastasis. WIN treatment resulted in the downregulation of cyclooxygenase-2 (COX-2) expression and decreased the phosphorylation of AKT, and inhibited EMT in SGC7901 human gastric adenocarcinoma cell line [117]. In lung cancer, tumor-associated macrophages (TAMs) has been reported to promote tumor growth by supporting the epithelium-to-mesenchymal transition (EMT). Treatment with JWH-015 inhibited EMT by regulating EGFR signaling targets such as ERK and STAT3 in NSCLC and A549 lung cells. In addition, JWH-015 in CALU-1 lung cancer cells was able to reverse their mesenchymal character. JWH-015 treatment of A549 lung cancer cells co-cultured with M2 polarized macrophages reduced the expression of FAK, VCAM1, and MMP2. In a syngeneic mouse model, JWH-015 decreased lung tumor lesions, tumor growth, and also inhibited macrophage recruitment and EMT to the tumor site via EGFR [78]. In endometrial cells (EC), treatment with THC impacted proliferative and migratory processes during impairment of cancer progression through motility by inhibiting EMT and downregulating matrix metalloproteinase 9 (MMP 9) gene expression in aggressive human EC cells [92]. Analysis of 157 tumor patients with non-small cell lung cancer (NSCLC) showed resistance to antitumor agents by mechanisms that involve the EMT. THC and CBD inhibited the proliferation and expression of EGFR in lung cancer cells, and CBD enforced the effect of THC. Finally, both cannabinoids reduced the in vitro migration of the A549, H460, and H1792 lung cancer cell lines. THC and CBD (alone and/or in combination) affected proliferation, EMT, and migration in vitro [118]. Yoshinaga T. et al. showed that the suppression of EMT by AM251, known to act as an antagonist of CB1R and an agonist of GRP55, was not mediated through either receptor. Microarray analyses show how AM251 reduced the expression of several EMT transcription factors such as SNAIL1, which is the key inducer of EMT, and the AP-1 transcription factors FOSB and JUNB. The synthetic lipid AM251 was able to suppress two important events associated with EMT—upregulation of collagen 1A1 (COL1A1) and downregulation of E-cadherin. AM251 inhibited the activation of SMAD2/3 and p38 MAPK, indicating that AM251 acted upstream of SMAD/p38 MAPK in the TGF-β signaling pathway [119]. Met-F-AEA inhibited the growth of adenocarcinoma breast cancer cells, MDA MB 231 cells. These data in vitro are confirmed also in an animal study showing that supplement therapy could inhibit breast tumor development in nude mice. Besides, we observed a much lower degree of breast tumor metastasis in mice treated with Met-F-AEA, which reduced protein levels of β-catenin by inhibiting the transcriptional activation of T Cell Factor (TCF) responsive element (a marker for β-catenin signaling). It has been shown that anandamide treatment upregulated epithelial markers, like E-cadherin with a concomitant decrease in protein levels of mesenchymal markers, including vimentin and SNAIL1. Indeed, anandamide treatment was able to inhibit induction of experimental epithelial-mesenchymal transition by exposure to adriamycin in MCF7 human breast cancer cell line [120] (illustrated in Figure 2 and Table 2).

## 5. Conclusions

Here, we described the anticancer action of the cannabinoids due to the modulation of intracellular signaling pathways implicated in cancer progression. A more intensive basic research, aimed at better identifying intracellular factors, molecular targets, and signaling mechanisms modulated by cannabinoids, can help us to progress in our understanding of the pharmacological basis of their anticancer action. Contradictory data in the literature about the efficacy of cannabinoids as anticancer agents highlight the gaps in our understanding of their mechanisms of action as well as in the physiological signaling pathways carried out by endocannabinoids. Future studies must be directed to resolve the conflicting evidence around the use of cannabinoids in clinical setting. The chemical diversity of cannabinoids may enrich the range of effects that they might induce. Enthusiastic data of animal-based in vivo studies show the antimetastatic and antiangiogenic effects of cannabinoids, which may serve as options for the currently used chemotherapy. Additionally, several preclinical investigations point to improved patient survival in combined administration of cannabinoids with chemotherapeutic drugs. Clinical trials have been carried out using a combined treatment of nabiximols and temozolomide in glioblastoma [11] and others are in progress; however, their use can currently be limited as adjuvant to chemotherapy. Scientists need to collect data regarding the risks and benefits of using cannabinoids for cancer patients due to the many legal and ethical issues associated with their use is currently limited. Future studies should further investigate the routes of administration and absorption of cannabinoids to improve their application in cancer therapy.

## Figures and Tables

**Figure 1 ijms-22-03680-f001:**
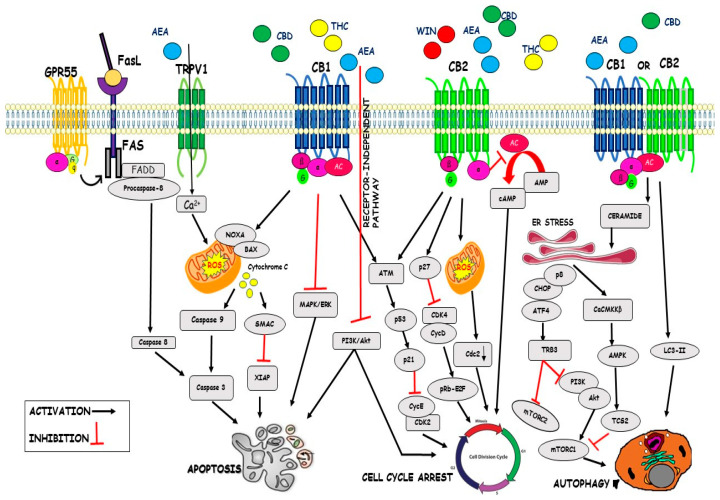
Main anticancer molecular pathways mediated by cannabinoids. Cannabinoids inhibit cell cycle progression (↓cyclin-CDK complexes, ↓cAMP, ↑ROS↓, PI3K/Akt), can induce apoptosis (ROS↑, caspase8-9↑, MAPK/ERK↓, PI3K/Akt↓) and autophagy (Ceramide↑, ER STRESS↑, mTORC1/2↓, LC3-II↑) by activation of cannabinoid receptors CB1 or CB2, but can be also induced by mechanisms independent of CB1 and CB2 receptors.

**Figure 2 ijms-22-03680-f002:**
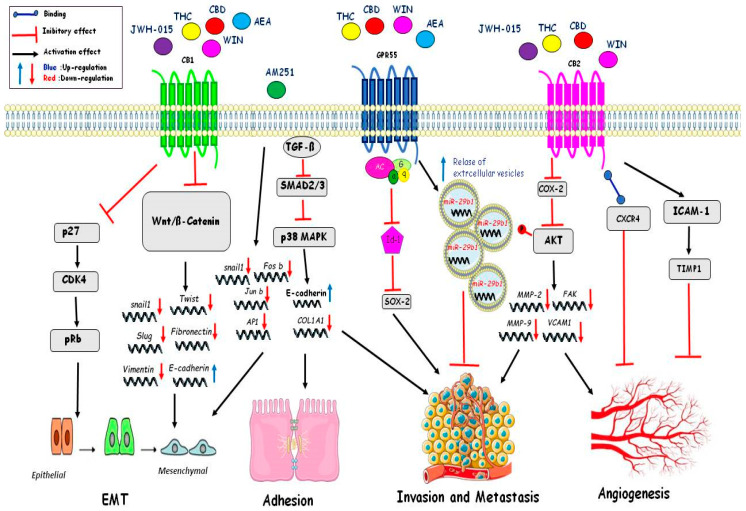
Schematic representation of the main mechanisms mediated by cannabinoids on angiogenesis, EMT, adhesion, invasion, and metastasis. Cannabinoids through the WNT/catenin pathway reduce the expression of genes implicated in EMT (snail1, twist, slug, and vimentin) and increase the expression of E-cadherin. Cannabinoids reduce angiogenesis and metastasis through release of extracellular vesicles containing miR-29b1 and inhibiting phosphorylation of AKT and expression of MMP2, MMP9, FAK, and VCAM1. Angiogenesis is inhibited by CB2 receptor binding CXCR4 receptor and by expression of TIMP1. Instead, AM251 inhibited adhesion and EMT, not through the CB receptor but via TGF-beta and P38 Map-kinase. Cannabinoids through GPR55 receptor decrease expression of Id-1 (helix-loop-helix transcription factor inhibitor of DNA binding 1) and consequentially of Sox 2 to inhibit angiogenesis and metastasis.

**Table 1 ijms-22-03680-t001:** Cannabinoids’ effects on signaling pathway of cell cycle, apoptosis, and autophagy.

	Mediators	Effects	References
**Cell Cycle**	WIN 55,212-2 CBD THC AEA	Inhibition of Cdk4/CycD Inhibition of Cdk2/CycE Downregulation of Cdc2 Inhibition of cAMP Inhibition of AKT pathway	[65] [87] [88] [90] [91]
**Apoptosis**	CBD C6 THC AEA metAEA	Inhibition of XIAP Downregulation Bcl-2 Upregulation Noxa Enhance ROS generation Upregulation of GRP78 Inhibition of ERK/AKT signalling Accumulation of Ceramide Activation of Raf1/ERK Downregulation AKT pathway Increase of Ca^2+^ influx Activation of Fas/FasL Activation of PPARγ	[92] [87] [92] [93] [94] [93] [95] [95] [91] [95] [96] [72]
**Autophagy**	THC-CBD THC AEA	Activation of CaCMKKβ Inactivation of mTORC1 Activation of LC3-II Inhibition of mTORC1 Downregulation SOCS3	[91] [91] [89] [100] [100]

**Table 2 ijms-22-03680-t002:** Cannabinoids’ effects on signaling pathway of EMT, adhesion, invasion, metastasis, and angiogenesis.

	MEDIATORS	EFFECTS	REF
**Epithelial Mesenchimal Transition (EMT)**	JWH-015 THC CBD WIN AEA	InhibitionWnt/ßcatenina:Downregulation of Snail, Twist, Slug, Vimentin, Fibronectin; Upregulation of E-Cadherin. Inhibition of p27/CDK4/pRb.	[120] [119]
	AM251	Downregulation of Snail1, Fos-b,Jun-b, Ap1.	[119]
**Adhesion**	AM251	TGFß inhibit Smad2/3-p38MAPK: Downregulation of COL1A1 e Upregulation of E-Cadherin	[119]
**Invasionand metastasis**	THC CBD WIN AEA	Inhibition Id1 and Sox2. Relase of extracellular vesicles with miR-29b1. Inhibition of Cox-2 and of phosphorylation AKT and inhibition of MMP2;FAK,MMP-9,VCAM1	[101] [104] [106]
**Angiogenesis**	JWH-015 THC CBD WIN	Inhibition of Cox-2 and phosphorylation of AKTand inhibition of MMP2; FAK; MMP-9; VCAM. Binding CXCR4 Activation of ICAM-1 and Timp1	[90,117] [92] [112]

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
