# Peer review of "Molecular Mechanism of Cannabinoids in Cancer Progression"

_ijms, 2021, doi:10.3390/ijms22073680_

Round 1

Reviewer 1 Report

The manuscript described by Pagano and coworkers presents nice review concerning action of cannabinoids on cancer proliferation, neovascularization, migration, invasion and metastasis.

The manuscript has 115 references, including as many as 73 of them from last decade.

I have only few comments:

lines 23-24: Abbreviations of AEA and 2-AG used for the first time should be explained. Then in line 24 Authors can use only abbreviation, similar in line 28.

line 147 and 558: The name of plant Cannabis sativa should be italic.

Editorial comments:

line 265: The chapter title should be bold according to MDPI template.

line 280: typo „G1 phas”, should be “G1 phase”.

In whole text there are many unnecessary hyphenated words e.g.: line 282 should be „negative”, line 283 should be „significantly”, line 295 should be “arresting”, line 321 should be “increased”, line 323 should be ”permeability”, line 324 should be “release”, line 331 should be „activation”… Please improve it.

line 309 and 502: Reference 115 should be in square brackets without “Ref.” before the number, should be [115], no [Ref.115] – please improve it. What is under the reference number 115 ?

line 418-419: The chapter title at the end of a paragraph „4. Mechanism of action of the cannabinoids on cancer progression” should be removed.

line 603: Number “4.4” at the end of sentence should be removed.

Author Response

Reviewer 1:

We would like to express our sincere thanks for your comments on our manuscript. We made suggested corrections for:

lines 23-24:  We did it.

line 147 and 558:  We did it.

Editorial comments:

line 265:  We did it

line 280: We did it

In whole text there are many unnecessary hyphenated words e.g.: line 282 should be „negative”, line 283 should be „significantly”, line 295 should be “arresting”, line 321 should be “increased”, line 323 should be ”permeability”, line 324 should be “release”, line 331 should be „activation”… Please improve it.

Answer to the reviewer: The hyphenated words depend on text formatting

lines 309 and 502:  We did it

line 418-419:  We removed it.

line 603:  We removed it

All corrections are highlighted in the manuscript.

Reviewer 2 Report

The authors review targeting the endocannabinoid system and cancer. Major comments: The diagrams are a good inclusion as they summarize a lot of the manuscript. In general there should be more citation of the literature as many statements are not cited. Throughout the manuscript, the authors need to be clear if the data is from tumors or tumor cell lines Minor Comments: Introduction: Define CB1 and CB2-line 20 Line 27-28 there needs to be a consistent naming of the receptors; are they CB receptors or CB1R CB2R? Line 33-43-where can you find CB1 and CB2-there are more organs than these, and there should be a citation How were non-CB1/2 receptors identified-line 46 Line 63-levels should be replaced with expression throughout Lines 67-68 require references Section 2.1 is largely a list with little integration and discussion of the differences observed. The data in cancer tissues should be separated from cell lines, as these may not reflect changes in the tissue Cannabinoid ligands There is no connection with the ECS expression in cancer cells. You should notify the reader, because the ECS is expressed in cancer cells, investigations have been made into the potential of modulating the ECS as a cancer therapeutic What specific anti-cancer effects? Line 281-CBD is defined earlier Figure 1 and 2-can these be made bigger?

Author Response

Reviewer 2

The authors review targeting the endocannabinoid system and cancer. Major comments: The diagrams are a good inclusion as they summarize a lot of the manuscript. In general, there should be more citations of the literature as many statements are not cited. Throughout the manuscript, the authors need to be clear if the data is from tumors or tumor cell lines Minor Comments: Introduction: Define CB1 and CB2-line 20 Line 27-28 there needs to be a consistent naming of the receptors; are they CB receptors or CB1R CB2R? Line 33-43-where can you find CB1 and CB2-there are more organs than these, and there should be a citation How were non-CB1/2 receptors identified-line 46 Line 63-levels should be replaced with expression throughout Lines 67-68 require references Section 2.1 is largely a list with little integration and discussion of the differences observed. The data in cancer tissues should be separated from cell lines, as these may not reflect changes in the tissue Cannabinoid ligands There is no connection with the ECS expression in cancer cells. You should notify the reader, because the ECS is expressed in cancer cells, investigations have been made into the potential of modulating the ECS as a cancer therapeutic What specific anti-cancer effects? Line 281-CBD is defined earlier Figure 1 and 2-can these be made bigger?

We would like to express our sincere thanks for your comments on our manuscript. We made suggested corrections for:

  1. a) The diagrams are a good inclusion as they summarize a lot of the manuscript. In general there should be more citation of the literature as many statements are not cited.

Answer to the reviewer: We have added 2 tables that resume the cannabinoid effects on signaling pathway described. More we have added new references in the manuscript.

  1. b) Throughout the manuscript, the authors need to be clear if the data is from tumors or tumor cell lines.

Answer to the reviewer: We specified the used models in the text.

Define CB1 and CB2-line 20 Line 27-28 there needs to be a consistent naming of the receptors; are they CB receptors or CB1R CB2R? Line 33-43-where can you find CB1 and CB2-there are more organs than these, and there should be a citation How were non-CB1/2 receptors identified-line 46 Line 63-levels should be replaced with expression throughout Lines 67-68 require references.

Answer: We corrected it

Section 2.1 is largely a list with little integration and discussion of the differences observed. The data in cancer tissues should be separated from cell lines, as these may not reflect changes in the tissue

Answer:  We briefly summarized the expression of cannabinoid receptors in  cancer because the review focuses on the effect of cannabinoids on signaling pathways of cell proliferation, migration, and angiogenesis. We separated cell lines from cancer tissues

Cannabinoid ligands There is no connection with the ECS expression in cancer cells. You should notify the reader, because the ECS is expressed in cancer cells, investigations have been made into the potential of modulating the ECS as a cancer therapeutic What specific anti-cancer effects?

Answer:  The anticancer effects of ECS we resumed in tables

Line 281-CBD is defined earlier

Answer: We did it

Figures 1 and 2-can these be made bigger?

Answer:  We tried to improve the figures. All corrections are highlighted in the manuscript.